# Quiescence modulates age-related changes in the functional capacity of highly proliferative canine lung mesenchymal stromal cell populations

Nakesha Agyapong⊙, Leslie Dominguez-Ortega⊙, Brian Macdonough⊙, Patrick Mulluso⊙, Sagar Patel⊙, Briti Prajapati⊙, Brian Saville⊙ID⊙, Andrew Shapiro⊙ID⊙, Ethan Trim⊙, Kara Battaglia, Jocelyn Herrera, Gianna Garifo-MacPartland, Dianne Newcomb, Latoya Okundaye, Heather Paglia, Julia Paxson⊙ID*

College of the Holy Cross, Worcester, Massachusetts, United States of America

⊙ These authors contributed equally to this work.
* jpaxson@holycross.edu

## Abstract

The functional capacity of highly proliferative cell populations changes with age. Here, we report that the proliferative capacity of canine lung mesenchymal stromal cells (LMSCs) declines with increasing age of the donor. However, other functional changes such as reduced autophagy, reduced migration/wound healing, increased production of reactive oxygen species, and increased senescence are not significantly altered with increasing age. Furthermore, transcriptomic profiling suggests minimal age-related changes. These data suggest that the reduced proliferative capacity of lung LMSCs isolated from aging donors may be associated with reversible cell cycle arrest (quiescence), rather than irreversible cell cycle arrest (senescence). Similar findings have been reported in other systems, including neural and muscle stem cells that are associated with low turnover-rate tissues.

## Introduction

Tissue homeostasis and repair in mature animals is mediated by populations of highly proliferative cells, including some types of mesenchymal stromal cells (MSCs). Lung-resident mesenchymal stromal cells (LMSCs) are important mediators of neonatal alveolar development, adult tissue repair, and are dysregulated in a variety of age-related interstitial lung pathologies [1]. Multiple studies have confirmed that these cells represent a heterogeneous population of highly proliferative multipotent progenitor cells that exhibit trilineage differentiation potential in culture and express characteristic mesenchymal cell surface proteins including CD29, CD44, CD90 [2–4]. In mice, LMSCs are critical in post-pneumoectomy regeneration (compensatory regrowth), with early proliferation of LMSCs associated with successful regeneration.

**Data availability statement:** All relevant data are within the manuscript and its Supporting Information files.

**Funding:** Research reported in this study was supported by the National Institute on Aging, National Institutes of Health, grant number R15 AG064558-01 to JP.

**Competing interests:** The authors have declared that no competing interests exist.

With increasing age, post-pneumonectomy regeneration declines and is associated with less proliferation of LMSCs [5,6].

Age-related declines in proliferative capacity have been observed both in-vivo and in-vitro in many different populations of stromal and stem cells [7,8]. Proliferative capacity in these cell populations is controlled by entry into and exit from the cell cycle and can be characterized as either reversible (quiescence) or irreversible (senescence) arrest [9,10]. The relative roles of quiescence and senescence in regulating cell proliferation in MSCs may vary depending on extrinsic signals received by the cells from the surrounding niche, as well as by cell-intrinsic factors [9]. Quiescence is thought to be a mechanism by which highly proliferative cell populations may be protected from replicative DNA damage and production of reactive oxygen species, allowing cellular integrity to be maintained in these populations [9,11]. Quiescence has been described in many different highly proliferative cell populations including hematopoietic, neural, muscle, and hair follicle stem cells, as well as in MSCs [9].

In contrast, senescence is triggered by a variety of stress signaling including oxidative stress, telomere shortening, DNA damage, mitochondrial damage, or inflammation [12]. In turn, senescent cells are often associated with altered cellular functions, including increased production of reactive oxygen species and increased autophagy [13,14]. Senescent cells remain viable, but they have altered metabolic activity, develop a pro-inflammatory senescence-associated secretory phenotype (SASP), and have dramatic changes in their transcriptomic profile [12,15,16]. While these mechanisms can contribute to aging and lack of tissue repair due to the irreversible cell cycle arrest, senescence is also potentially an anti-oncogenic mechanism to avoid allowing genomically unstable or damaged cells to continue proliferating [12].

Understanding the relative roles of quiescence and senescence in regulating age-related declines in the proliferative capacity of MSCs has important implications for potentially re-activating these cells later in life [9,11]. In many studies, age-related declines in stromal and stem cell proliferation have been linked to increased senescence, suggesting that the associated cell cycle arrest is irreversible [17,18]. However, some studies have documented populations of stem cells that experience age-related increases in quiescence but not senescence, suggesting that the cell cycle arrest in these cells may be reversible [11,19]. Understanding how different populations of highly proliferative cells respond to aging is critical to developing therapies to combat age-related diseases, such as chronic interstitial lung diseases [1]. For example, it is possible that populations of cells that contribute to repair in low-turnover tissues such as brain, muscle and lung have different aging mechanisms than cell populations in tissues with high cell turnover such as epithelial tissues [8].

Companion dogs are well-suited to the study of LMSC aging since they experience many of the same aging characteristics as humans, are relatively long-lived, and are exposed to many of the same environmental factors as their human owners [20,21]. In addition, breed and size differences among companion dogs allow us to

investigate differences in aging characteristics in different populations of genetically diverse animals [20,21]. Characteristics of canine adipose-derived MSCs isolated from young versus aged dogs have been reported, but only in relation to their suitability for therapeutic applications. Lee et al reported that canine MSCs derived from adipose tissue had age-related changes consistent with reduced reparative function including lower population doubling times, expression of characteristic MSC surface markers, and reduced differentiation potential [22]. However, the gene expression profiling in this study showed very little difference in expression patterns between young and old dogs, suggesting that there may not be significant age-related changes at a molecular level. Two additional studies have reported the effects of age on canine adipose-derived MSCs, but with very limited characterizations of the molecular mechanisms driving the observed age-related declines [23,24].

In this study, we have used LMSCs isolated from companion dogs to identify how the function of these highly proliferative cells changes with the relative age (percent predicted lifespan) of the donor. Our data indicate that cell proliferation declines with age, but that many other cellular functions remain statistically unchanged with age, including autophagy, ROS production, and migration capacity. Furthermore, the transcriptomic profile of these cell populations is similar across donors of different ages. Finally, LMSCs isolated from older dogs do not show any significant increase in senescence, suggesting that the predominant mechanism for cell cycle arrest in these cell populations is likely quiescence. This is consistent with observations made in highly proliferative cell populations from tissues with low turnover rates and may suggest a similar aging mechanism among these cell types [8].

## Materials and methods

### Animals and LMSC isolation methods

Postmortem distal lung tissue samples were obtained from companion dogs of various breeds and ages with no observable lung pathology (Table 1). Written informed consent was obtained from each owner at the time of euthanasia. Since dog lifespan can be correlated with the breed and/or size of the dog, each dog was given a calculated % predicted lifespan to normalize the ages of the donors across the different breeds that were represented. Guidelines from the American Kennel Club were used to calculate predicted lifespan for each animal (Table 1). Two groups of dogs were created for analysis: <50% of predicted lifespan and > 50% of predicted lifespan.

For each dog, the lung samples were collected in Dulbecco's Modified Eagle Medium (DMEM, Gibco Thermofisher # 11965118) with 5x antibiotics/antimycotics (A/A) (Gibco Thermofisher # 15240062) (DMEM 5x A/A) within 4 hours of euthanasia. A previously established explant-outgrowth method was used to isolate LMSCs [5,25]. Briefly, the lung tissue was finely minced in DMEM 5xA/A until the average tissue piece size was 1−2 mm$^3$. These small tissue pieces were placed on 60 mm plastic cell culture plates with DMEM 5xA/A + 15% FBS just covering the pieces (about 1.5 ml) AND incubated at 37°C with 5% CO2. The following day, the media was replaced with 4 ml of DMEM with 15% FBS, 1% A/A, and 1% glutamine. After roughly 5−10 days (depending on the age of the donor and the rapidity of cell migration and proliferation), LMSCs could be observed on the cell culture plates and the tissue pieces were removed. When the cells reached 80% confluence, they were trypsinized (0.25% TrypLE, Thermo Fisher cat# 12563-011), lifted and replated to P1, then cryopreserved for storage in 60% FBS, 30% DMEM, and 10% DMSO. All experiments were conducted after cells have been thawed and passaged until P5. All incubation periods unless otherwise stated occurred in a 37°C, 5% CO2 incubator. DMEM media with 15% fetal bovine serum (FBS, Gibco Thermofisher # 16000044), 1% antibiotics (A/A), and 1% glutamate (Gibco Thermofisher # A2916801) was used as growth media.

### Characterization of cell surface markers

LMSCs to be analyzed for the presence or absence of surface protein markers were grown to about 80% confluence at P5, then washed three times with non-supplemented DMEM medium. The cells were then trypisinized and washed three times with DMEM containing 1% FBS (flow media) to remove the TrypLE. After washing, the cells were stained

**Table 1. Characteristics of the donor dogs used in this study.**

| % predicted lifespan | Age | Breed | Sex (F/M) |
|---|---|---|---|
| 0.01 | 4 weeks | German shepherd | F |
| 0.01 | 4 weeks | Bernese mountain | F |
| 0.02 | 11 weeks | Maltese | F |
| 0.16 | 2 years | Hound cross | M |
| 0.19 | 2.5 years | Pitbull | F |
| 0.29 | 4 years | Daschund | F |
| 0.33 | 4 years | Labrador cross | F |
| 0.42 | 5 years | Labrador | M |
| 0.54 | 7 years | English springer spaniel | F |
| 0.59 | 8 years | Australian Shepherd | M |
| 0.74 | 10 years | Cavalier King Charles Spaniel | M |
| 0.77 | 10 years | Pitbull | F |
| 0.81 | 11 years | Miniature Schnauzer | M |
| 0.87 | 10 years | Greyhound | M |
| 0.93 | 14 years | West Highland Terrier | M |
| 1 | 11 years | Golden Retriever | F |
| 1 | 14 years | Pug | M |
| 1.03 | 15.5 years | Rat Terrier | F |
| 1.18 | 10 years | German Shepherd | M |
| 1.36 | 15 years | Golden Retriever | M |

with the appropriate antibody to properly characterize their surface protein expression. Cells were separately incubated with FITC-conjugated anti-CD29 (Invitrogen Ref#MA1-19566), FITC-conjugated anti-CD34 (Bio-Rad Ref#MCA4211F), FITC-conjugated anti-CD44 (Bio-Rad Ref#MCA1041A488), FITC-conjugated anti-CD45 (Bio-Rad Ref#MCA1042F), or PE-conjugated anti-CD90 (Invitrogen Ref#17-5900-42) for one hour in 2 °C. To remove unbound antibodies, cells were washed three times with flow media after the incubation period. Fluorescence was compared to an unstained control sample and all data were analyzed using FlowJo™ software (Tree Star Inc., Ashland, OR).

### Population Doubling Time (PDT) assay

PDT assays were conducted in triplicate on 60 mm plates using P5 cells. Cells were plated at 56,000 cells per plate (2,000 cells/cm2). Plating time and date were noted. Media was changed every 2 days. Once plates appeared to be ~80% confluent, cells were lifted and counted. Time and date of lifting were also noted. Population doubling time was calculated using the following equation: $PDT = t*(\ln(2)/(\ln(Xe/Xb))$. In this equation, t is the time it took for the plates to reach ~80% confluence, which is the time between plating and lifting. This number is normally rounded to the closest 15 minutes (like 6 PM, 6:15 PM, 6:30 PM, etc.). $Xe$ is the final cell count and $Xb$ is the initial cell count (56,000 cells). The PDT for each of the plates was noted and the mean of the three were used as the average PDT for that particular dog and passage. Final PDTs are presented in hours.

### Colony-Forming Units (CFU) assay

CFU assays were conducted in triplicate on 60 mm plates using P5 cells. Cells were plated at 1,400 cells per plate. Plating time and date were noted. Media was changed every 2 days. 8 days after plating, each plate was stained using 0.1% crystal violet staining solution for 1–2 hours. Each plate was then rinsed with deionized water and the total

number of colonies for each was counted using a light microscope. The number of colonies per plate were noted and the mean of the three were used as the average CFU for that particular dog and passage. Final CFU data is presented in percentage colonies formed, which is found by dividing the average number of colonies formed by the total initial cell count.

### EdU Incorporation assay

EdU assays were conducted in triplicate on 60 mm plates using P5 cells. Cells were plated at 56,000 cells per plate (2,000 cells/cm2). Plating time and date were noted. Media was changed every 2 days. Experiments lasted 72 hours total. 52 hours into the experiment, 4 µL of EdU (5-ethynyl-2'-deoxyuridine) solution was added into the existing 4 mL of media on each plate (a 1:1000 dilution). 20 hours later, we began the EdU protocol using the

Invitrogen™ Click-iT™ Plus EdU Cell Proliferation Kit for Imaging, Alexa Fluor™ (#C10637). Protocol for fixation and expression provided in the kit was followed. Plates were counted using a standard fluorescent microscope. Five to ten images of each plate were taken. EdU+ cells were those that stained green, indicating EdU incorporation. Total cell count was quantified using DAPI staining. The EdU count for each of the plates was presented as the total number of EdU+ cells divided by the total number of cells (total number of DAPI-stained cells). The mean of the three were used as the average EdU for that particular dog and passage. Final EdU data is presented in percentage of EdU+ cells.

### Cellular autophagy assay

Cellular autophagy assays conducted in triplicate on 60 mm plates on P5 cells at 80% confluence using the Enzo Life Sciences CYTO-ID Autophagy detection kit 2.0 (#NC1500143). Cells were then incubated for 30 mins, then trypsinized, lifted, resuspended in PBS + 1% FBS and analyzed by flow cytometry. All samples (at least 50,000 cells in each sample) were analyzed using the Accuri BD Accuri C6 Flow Cytometer. Prior to analyzing samples, all were checked for cell clumping under a compound microscope.

### Reactive oxygen species (ROS) assay

ROS assays were conducted in triplicate on 60 mm plates on P5 cells at 80% confluence using the ThermoFisher CellROX™ Green Flow Cytometry Assay Kit (#C10492). Cells were then incubated for one hour, then trypsinized, lifted, resuspended in PBS + 1% FBS and analyzed by flow cytometry. All samples (at least 50,000 cells in each sample) were analyzed using the Accuri BD Accuri C6 Flow Cytometer. Prior to analyzing samples, all were checked for cell clumping under a compound microscope.

### Scratch assays

Scratch assays were performed as previously described [26]. Briefly, the assays were conducted in triplicates on 60 mm plates using P4 cells at 95–100% confluence. Two vertical scratches are created using a 200ul pipette tip equidistant from each other along the length of the plate. Images were taken along the length of each scratch using an inverted microscope and a SwiftImaging microscope camera, after which the plates were returned to the incubator. After 9 hours, repeat images were taken. Images were then analyzed using the MRI_Wound_Healing_Tool from the software ImageJ. ImageJ was used to calculate the total area of the scratch before and after migration. Subtracting the two values generated measurements of the total migration area or scratch closure (measured in pixels).

### Senescence assays

Two different senescence assays were performed to confirm our observations. In both assays, senescence was measured by production of SA-BGal on cells at P5 at 60–70% confluence (log growth phase). A flow cytometry-based senescence

assay was performed using the Invitrogen CellEvent™ Senescence Green Flow Cytometry Assay Kit (#CD10841). Following the incubation, cells were resuspended in 1% FBS/PBS for flow cytometry analysis. All samples (at least 50,000 cells in each sample) were analyzed using the Accuri BD Accuri C6 Flow Cytometer. Prior to analyzing samples, all were checked for cell clumping under a compound microscope.

In addition to the flow cytometry-based senescence assay, we also used a colorimetric senescence assay for confirm our results. Cells were assayed in triplicate at P5 at 60–70% confluence using the Cell Signaling Technology Senescence β-Galactosidase Staining Kit (#9860). The cells were washed once with phosphate-buffered saline (PBS) and then fixed using 0.5% glutaraldehyde for 10 minutes at room temperature. After fixation, the cells were incubated and sealed with parafilm with the staining solution at 37°C for 24 hours. Following the incubation period, the cells were observed under a light microscope, and images were captured. Senescent cells were identified by the presence of blue staining, indicative of β-galactosidase activity. To quantify the senescent cells, three researchers independently counted the senescent cells and averaged the results.

## Transcriptomic profiling

RNA was prepared for high through-put sequencing from 1 million cells at P5 at 80% confluence using the Zymo Research Direct-zolTM RNA MiniPrep w/TRI Reagent (#R2051). RNA quality was confirmed using a Nanodrop analyzer. Poly(A) RNA selection, cDNA library construction, sequencing (illumina HiSeq 2x150bp) and bioinformatics was performed by Azenta Life Sciences. The bioinformatics analysis workflow included evaluation of sequence quality, then trimming the sequence reads to remove possible adapter sequences and nucleotides with poor quality using Trimmomatic v.0.36. The trimmed reads were mapped to the canis_familiaris_ERCC reference genome available on ENSEMBL using the STAR aligner v.2.5.2b. The STAR aligner is a splice aligner that detects splice junctions and incorporates them to help align the entire read sequences. BAM files were generated as a result of this step. Unique gene hit counts were calculated by using featureCounts from the Subread package v.1.5.2. The hit counts were summarized and reported using the gene_id feature in the annotation file. Only unique reads that fell within exon regions were counted. After extraction of gene hit counts, the gene hit counts table was used for downstream differential expression analysis. Using DESeq2, a comparison of gene expression between the customer-defined groups of samples was performed. The Wald test was used to generate p-values and log2 fold changes. Genes with an adjusted p-value < 0.05 and absolute log2 fold change > 1 were called as differentially expressed genes for each comparison.

## Data analysis

Flow cytometry was analyzed using the FlowJo software and data values including mean and median fluorescence were recorded in Microsoft Excel. Statistical analysis was performed using Microsoft Excel and the data was visualized in GraphPad. For all data, statistical analyses were performed using an unpaired t-test with a Welch's correction. Normality assumptions were confirmed using the Shapiro-Wilk test. Equal variance assumptions were confirmed using F-tests. To evaluate the flow cytometry data collected from the Autophagy, CellROX Green and β-Galactosidase Senescence assays, median fluorescence intensity (MFI) values were standardized relative to untreated control cells from the same cell line by calculating the fold-change in MFI as described by the following formula: MFI (sample)/MFI (control). Three-way analysis of variance (ANOVA) tests with treatment/time point, dog sex, and LMSC age category (< 50% or > 50%) as factors were used to analyze the fold-change in MFI across for each individual assay. Interaction terms were included in each model to assess the impact of dog sex on fold-change in MFI of LMSCs from different age categories and the impact of LMSC age category on the fold-change in MFI of LMSCs under different treatments/time points. Post hoc Tukey HSD tests were used for pairwise comparisons. All analyses were performed in Excel or R version 4.2.2 (2022). Values p < 0.05 were considered significant in all analyzes.

## Results

### Characterization of primary lung MSC lines isolated from donor companion dogs

Lung samples were obtained from a total of 20 companion dogs ranging in age from 4 weeks to 15 years. Using predicted lifespan (American Kennel Club), dogs were classified as either young (yLMSCs < 50% predicted lifespan) or aged (aLM-SCs > 50% predicted lifespan). There were 8 dogs in the young group (and more females) and 12 dogs in the old group (with more males).

LMSCs were isolated from each dog using the explant-outgrowth method [5,25] (Fig 1A). We speculate that this may generate highly proliferative cell populations that are more likely to retain in-vivo properties than using collagenase digestion [27]. For each cell population, cell surface marker characterization was performed (Fig 1B). Our data are consistent with previous studies reporting that LMSCs isolated using this method generally represent a heterogenous population of cells that are CD34-, CD45-, CD29 + , CD44 + , CD90+ [2–4] (Fig 1B).

### The proliferative capacity of lung MSCs declines with age

A common characteristic of highly proliferative stem and stromal cell populations is an age-related decline in proliferative capacity [7,8]. Cellular proliferative can be measured in a variety of ways that can highlight either the proliferative capacity of individual cells, or the proliferation characteristics of cell populations. Characterization of age-related lung MSC proliferative capacity was performed using a combination of three different assays. Population doubling time (PDT), generation of colony forming units (CFUs), and incorporation of EdU (5-ethynyl-2'-deoxyuridine) during DNA replication and cell division (Fig 2).

**Population doubling time.** Population doubling time (PDT) assays determine how quickly a population of cell can double in size. Cells are plated at the same starting number and recounted after the cells reach ~80% confluence (with the time recorded in hours). The initial and final counts are used to determine the amount of time that it took for the

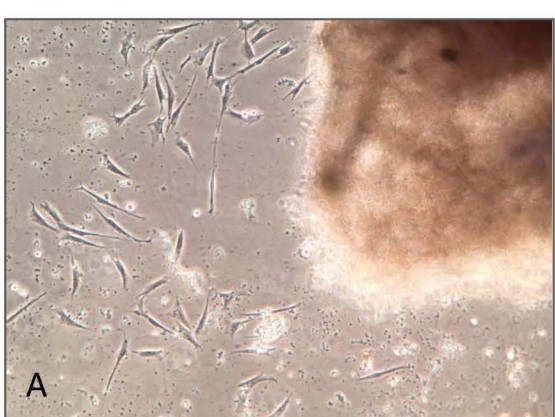
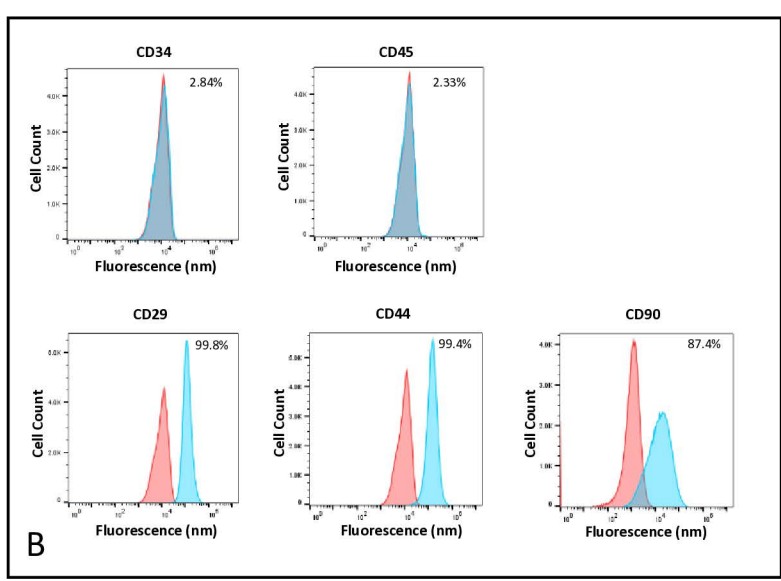

**Fig 1. Characteristics of yLMSCs and aLMSCs. A.** LMSCs were isolated using explant outgrowth. Individual cells can be observed migrating out from the minced tissue sample. **B.** Cell surface markers (Positive: CD29, CD44, CD90; Negative: CD34, CD45) observed by flow cytometry analysis at P5 indicate the presence of a homogeneous canine LMSC population. Blue curves indicate control cell populations while red curves indicate experimental cell populations stained for the marker of interest (n = 100,000 events in each sample in a representative cell line).

population to double. PDT assays were conducted at P5. Using Single Factor Analysis of Variance, we found that yLMSCs have significantly shorter population doubling times than aLMSCs (Fig 2A). The data indicated that yLMSCs on average had a population doubling time of 23 hours, while aLMSCs on average doubled their population after 38 hours.

**Colony-forming units.**  Colony-forming unit assays determine the ability of single cells to form colonies of more than 50 cells (considered highly proliferative). Cells are plated at very low density and allowed to grow for a predetermined length of time. During this time, only the most proliferative cells can divide sufficiently to for a visible colony, CFU assays were conducted at P5. Using Single Factor Analysis of Variance, we found a significant difference between the number of cells capable of forming colonies formed in yLMSC populations compared to the aLMSC populations (Fig 2B).

**EdU incorporation.**  EdU incorporation is another measure of cell proliferation that labels actively dividing cells through incorporation of EdU into replicating DNA. Any cell that has divided will be tagged. Cells in this assay were exposed to EdU for 20 hours before analysis. Using Single Factor Analysis of Variance, we found that yLMSCs have a significantly higher percentage of EdU+ cells than aLMSCs (Fig 2C).

### The cellular functions of LMSCs isolated from young and old donors are similar

In addition to cell proliferation, we also examined several cellular functions that are commonly reported to demonstrate age-related changes, including autophagy capacity, ROS production, and migration capacity (Fig 3).

**Cellular autophagy.**  Cellular autophagy is a highly conserved lysosome-mediated cellular process that is used in all cells to degrade and remove a variety of defective cellular components such as organelles and protein aggregates [28]. During cellular aging, there have been reports of altered autophagy although the exact changes that are described vary between different cell populations [28]. Using a flow cytometry-based autophagy detection kit, we quantified the levels of cellular autophagy across our populations of yLMSCs and aLMSCs. However, there was no significant change in the level of autophagy observed across all cell lines, with variability observed in both yLMSC and aLMSC cell lines (Fig 3A).

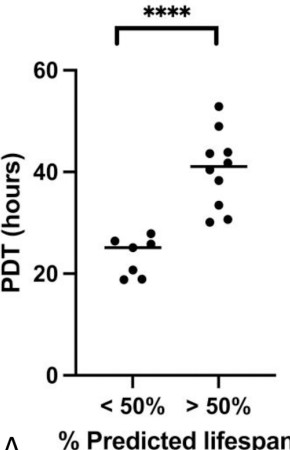 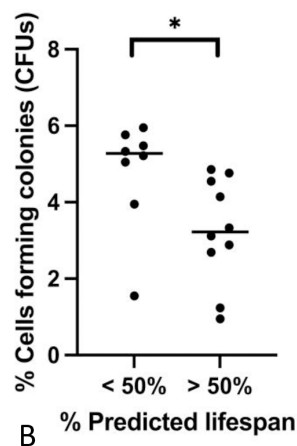 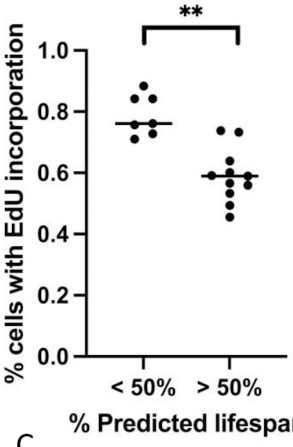

**Fig 2. Comparison of proliferation capacity in yLMSCs vs aLMSCs. A.** Population doubling time (PDT) assays were performed and the data were analyzed using predicted lifespan to group animals by age. Statistical analysis was performed using an unpaired t-test with a Welch's correction. **** indicates a p-value of < 0.0001. < 50% lifespan: mean 23.4hrs ± 3.8; < 50% lifespan: mean 40.4hrs ± 7.5. **B.** Colony forming unit (CFU) assays were performed and the data were analyzed using predicted lifespan to group animals by age. Statistical analysis was performed using an unpaired t-test with a Welch's correction. * indicates a p-value of < 0.05. < 50% lifespan: mean 4.8% ± 1.4; < 50% lifespan: mean 3.3% ± 1.4. **C.** EdU incorporation assays were performed and the data were analyzed using predicted lifespan to group animals by age. Statistical analysis was performed using an unpaired t-test with a Welch's correction. * indicates a p-value of < 0.01. < 50% lifespan: mean 0.8% ± 0.1; < 50% lifespan: mean 0.6% ± 0.1.

**Reactive oxygen species production and/or accumulation.** Production and/or accumulation of reactive oxygen species (ROS) is a process that has been described in some populations of aging MSCs [17]. Using a flow cytometry-based ROS detection kit, we quantified the total levels of ROS species across our populations of yLMSCs and aLMSCs. However, there was no significant change in the level of ROS accumulation and/or production observed across all cell lines, with variability observed in both yLMSC and aLMSC cell lines (Fig 3B).

**Migration capacity.** To further elucidate the reparative capacity and functional phenotyping of LMSCs, we used an optimized scratch migration assay protocol previously described [26]. The assay determines the capacity of cells to migrate across a cell culture plate surface over a given timeframe. Once again, there were no significant differences in the capacity of aLMSCs to migrate (scratch closure) compared to yLMSCs (Fig 3C).

## The amount of senescence present in populations of LMSCs isolated from young and old donors is similar

Many studies have documented age-related increases in cellular senescence, suggesting that reduced proliferation in these cell populations is associated with irreversible cell cycle arrest [17,18]. Age-related senescence can be quantified through the increased production of senescence-associated beta-galactocidase (SA-BGal). In an attempt to discern age-related differences in cellular senescence, we quantified the levels of senescence in populations of both yLMSC and aLMSC cell lines using two different methods – a flow cytometry-based assay and a colorimetric assay. However, we were unable to identify a significant difference in SA-βgal expression between the two age groups at P5 using either assay (Fig 4A, 4C, 4E). We also plotted the association between SA-βgal expression and donor dog age at P5 to test whether there was a correlation between senescence and age at different points across the lifespan of the donor dogs. We found a slight, linear, positive relationship between the variables, however, a simple linear regression showed that neither of these relationships were a significant correlation (Fig 4B).

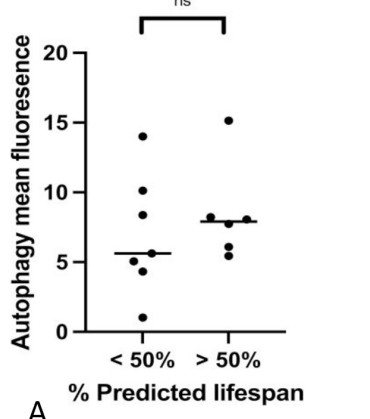
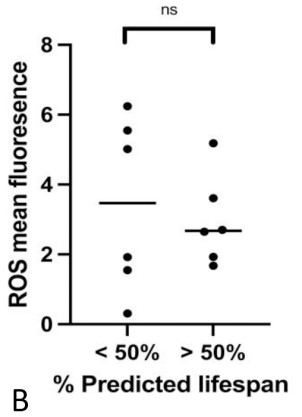
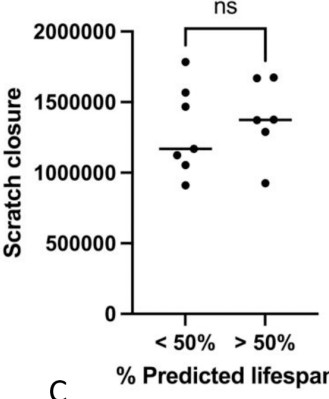

**Fig 3. Comparison of autophagy, presence of reactive oxygen species (ROS) and migration in scratch closure in yLMSCs vs aLMSCs.** A. autophagy assays were performed and the data were analyzed using predicted lifespan to group animals by age. Statistical analysis was performed using an unpaired t-test with a Welch's correction. ns indicates a p-value of > 0.05. <50% lifespan: mean 7.0 MFI±4.3; <50% lifespan: mean 8.5 MFI±3.5. **B.** ROS assays were performed to assess the presence of reactive oxygen species and the data were analyzed using predicted lifespan to group animals by age. Statistical analysis was performed using an unpaired t-test with a Welch's correction. Ns indicates a p-value of > 0.05. <50% lifespan: mean 3.4 MFI±2.5; <50% lifespan: mean 3.0 MFI±1.3. C. scratch assays were performed to assess the ability of cells to migrate and close the scratch and the data were analyzed using predicted lifespan to group animals by age. Statistical analysis was performed using an unpaired t-test with a Welch's correction. ns indicates a p-value of > 0.05. <50% lifespan: mean 129664.4 pixels±314790.2; <50% lifespan: mean 1384730.3 pixels±276980.8.

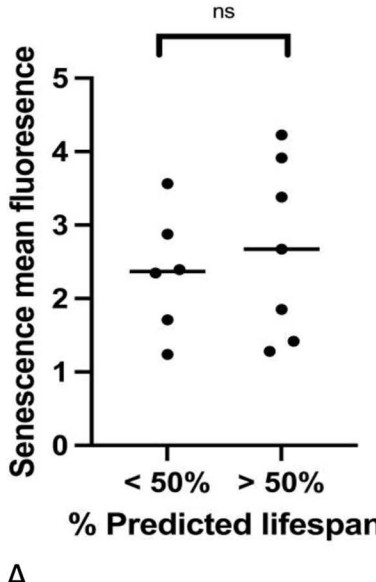

A

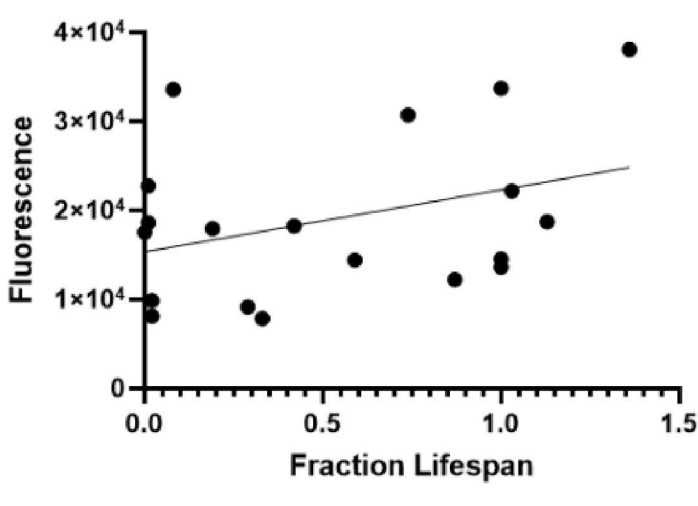

B

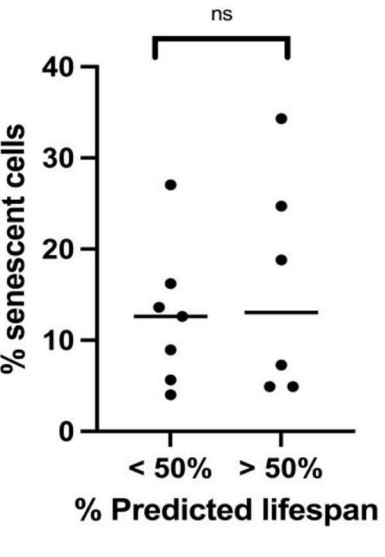

C

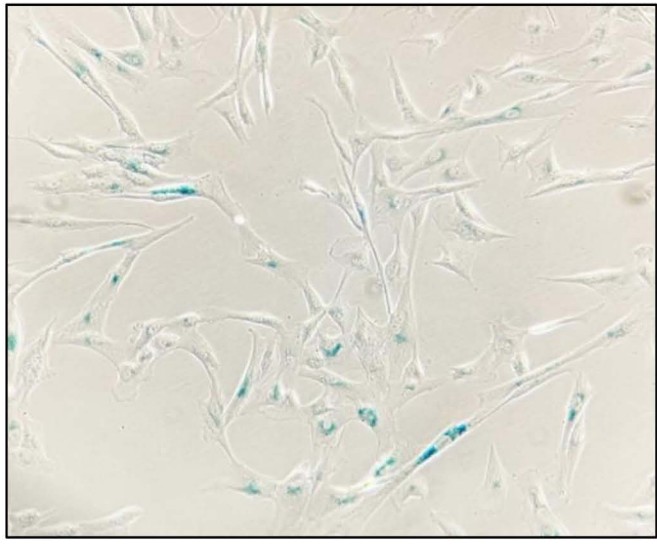

D

**Fig 4. Characterization of senescent cells in yLMSCs compared to aLMSCs. A.** SA-βgal expression in LMSCs at P5 was measured using the Invitrogen CellEvent™ Senescence Green Flow Cytometry Assay Kit. Statistical analysis was performed using an unpaired t-test with a Welch's correction. ns indicates a p-value of > 0.05. <50% lifespan: mean 2.4 MFI±0.8; <50% lifespan: mean 2.7±1.2. **B.** SA-βgal expression in LMSCs from this assay were also plotted against the fraction of average lifespan lived of the donor dog. There appears to be a slight, linear, positive correlation between the age of the donor dog and the level of SA-βgal expression. However, a linear regression failed to identify a significant correlation between the variables (p = 0.09). **C.** SA-βgal expression in LMSCs at P5 was also quantified using a standard colometric assay. Statistical analysis was performed using an unpaired t-test with a Welch's correction. ns indicates a p-value of > 0.05. <50% lifespan: mean 12.6%±7.7; <50% lifespan: mean 15.9±12.2. **D.** The colorimetric assay was performed using triplicate plates read independently by three different researchers to ensure standardization of the observed positive blue cells.

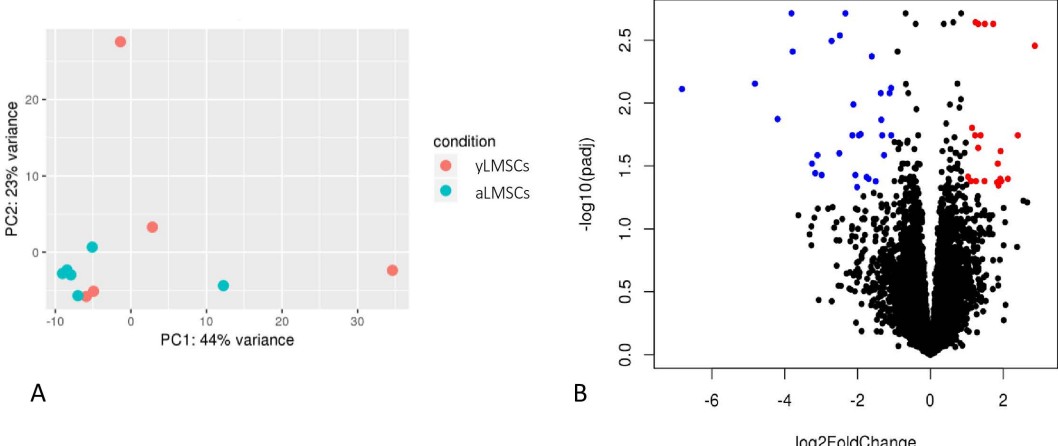

**Fig 5. Transcriptomic profiling in yLMSCs compared to aLMSCs. A.** Principle component analysis revealed similarity within and between the two groups. **B.** A volcano plot also illustrates a comparison of the global transcriptional change across the groups compared. All the genes are plotted and each data point represents a gene. The log2 fold change of each gene is represented on the x-axis and the log10 of its adjusted p-value is on the y-axis. Genes with an adjusted p-value less than 0.05 and a log2 fold change greater than 1 are indicated by red dots. These represent upregulated genes in aLMSCs compared to yLMSCs. Genes with an adjusted p-value less than 0.05 and a log2 fold change less than −1 are indicated by blue dots. These represent downregulated genes in aLMSCs compared to yLMSCs.

### The transcriptomic profiles of LMSCs isolated from young and old donors are similar

Aging cell populations with increased levels of senescent cells can often be identified by significant changes in their transcriptomic profile, associated with the development of a senescence-associated secretory phenotype (SASP) [15]. Therefore, comparison of transcriptomic profiles in yLMSCs compared to aLMSCs were performed using high throughput RNAseq. As evidenced by both the principle component analysis (PCA) and associated volcano plot, there are very few differences in the transcriptomic profiles between yLMSCs and aLMSCs, despite the genetically heterogeneous population of donor animals (Fig 5).

### Discussion

Understanding cellular aging in populations of highly proliferative cells such as MSCs that are physiologically important for repair and tissue homeostasis is a critical question in aging research. The traditional dogma for aging in many highly proliferative stem and stromal cell populations stipulates that there are age-related reductions in cell proliferation, reductions in cellular functions (such as increased ROS production), increased cellular senescence, and significant changes in the transcriptomic profile at a population level [17,29]. However, recently several studies have reported an alternative aging scenario in muscle and neural stem cell populations where increased age is associated with reduced cell proliferation, but without accompanying alterations in cell function or alterations in transcriptomic profiling across the cell populations [8,11]. In these cell populations, researchers suggest that the reduced proliferative capacity is associated with age-related increases in quiescence but not senescence, suggesting that the cell cycle arrest in these cells may be reversible [11,19]. Our characterization of canine lung MSCs suggests a similar trend, with age-related declines in cell proliferation, but no significant alterations in other cellular functions including autophagy, ROS production, migration, and senescence, or large-scale changes in the transcriptomic profile of these cell populations. Similar to the work in both muscle and neural stem cells, these data suggest that age-related declines in proliferative capacity are not related to increased cellular senescence, which would suggest that instead, these cells are being sequestered from the cell cycle through quiescence.

Since these data are consistent with observations made in other highly proliferative cell populations from tissue with low turnover rates, our study adds to the evidence suggesting that there may a similar aging mechanism among these cell populations [8].

Using companion dogs to study complex mammalian aging provides an ideal mammalian model system in which to further our understanding. Companion dogs provide several advantages over traditional rodent models of mammalian aging, including similarities both in life stages and environmental stressors that are more comparable to humans. Different breeds of companion dogs are also more phenotypically and genetically diverse compared to traditional rodent model systems [20,21]. In addition, some studies suggest that lab rodents represent negative outliers with respect to their expected lifespan as a function of their body size and onset of sexual maturity, with important variations in their typical aging patterns when compared to humans [30]. Aging in humans and other large predators such as dogs fits a much different survivorship curve compared to rodents (type I versus type II), which has also likely led to the evolution of different aging strategies to enable the optimal survival strategies for each survival type [30]. In the lung, previous studies suggest that post-pneumonectomy lung regrowth may occur in rodents through a greater proportion of their lifespan than humans [2], which may be consistent with more active LMSC populations in these animals compared to long-lived species such as humans and dogs. Therefore, studying aging in stem and stromal cell populations from particularly low-turnover tissues may be confounded by marked differences in the activity of regenerative cell populations and the associated reparative capacities of rodents [31].

In this study, we report the cellular characteristics of LMSCs isolated from 20 companion dogs of different ages. These dogs represent a phenotypically and genetically diverse pool of animals, with different environmental exposures, similar to their human companions. Although our study was limited by unequal sex distributions across the two groups, previous studies indicate that unequal sex distribution may not be as critical in canines as humans, since large-scale studies of sex differences in dogs have revealed fewer differences [32]. Analysis of our data supports this, with no sex or breed differences observed in any of our assays. While the heterogeneity of phenotypes, genotypes and potential environmental exposures may make it more difficult to observe patterns when comparing dogs of many different breeds, it may also increase the significance of any large-scale trends that emerge, suggesting more robust conserved mechanisms [20]. In this study, we performed a series of assays designed to examine how the function of isolated LMSCs changes with increasing donor age. This included assessment of surface protein markers, which confirmed that the surface marker characteristics of mesenchymal stromal cells used in this study are similar to widely-established characteristics described in other MSC populations [2–4], as well as assessment of proliferative capacity (including proliferation doubling time, colony forming units, and EdU incorporation). In addition, we report the effect of donor age on the capacity of LMSCs to mediate cellular levels of reactive oxygen species, to perform cellular autophagy, and to migrate in cell culture. We also report a comparison of two different assays quantifying the levels of senescent cells within these LMSC populations. Finally, we report trends in the transcriptomic profiles present in these cell populations. Human MSCs have been shown to have reduced proliferative capacity due to increasing donor age [33]. Autophagy, a cellular process for degrading damaged or necessary cellular components, has also been shown to decrease in aged MSCs [34]. Additionally, high ROS-levels have also been implicated in driving MSC aging and functional decline [35]. Furthermore, the aforementioned cellular mechanisms may contribute to cellular senescence or the arrest of the cell cycle through a complex interplay between DNA damage, epigenetic alterations, and functional decline [17]. Thus, identifying primary drivers of aging in MSCs may enable effective interventions to treat age-related disease pathologies.

Our data indicate that LMSC populations isolated from younger donors had a significantly higher capacity to proliferate than LMSCs isolated from older donors. LMSCs from younger donors take a significantly shorter time to double their population, they are more cells capable of forming colonies, and have more EdU+ cells. However, we did not observe significant age-related differences in other cellular functions. Importantly, although we did observe variability

in the level of senescent cells present in young and aged LMSC populations, we did not see any significant trends of increased levels of senescent cells with age. Senescence is induced by damage to DNA, proteins, and other macro-molecules over time by replicative stress and environmental factors such as reactive oxygen species [7,36]. Senescent cells also often demonstrate significant changes in their transcriptomic profiles due to the emergence of the senescence-associated secretory phenotype (SASP). Cells exhibiting an SASP undergo changes in gene regulation that promote the secretion of inflammatory cytokines, chemokines, growth factors, and proteases [37]. Age-related senescence can be measured by the presence of senescence-associated beta galactosidase (SA-βgal). SA-βgal is a lysosomal enzyme which is produced by the cell under normal conditions, but is significantly overexpressed in the senescent phenotype [38]. We used two different assays to measure the levels of SA-βgal in our young and aged LMSC cell populations, but neither assay demonstrated a significant age-related increase in SA-βgal+ cells. Consistent with this, we also could not demonstrate significant age-related changes in the transcriptomic profile of LMSCs that might be consistent with the development of a SASP phenotype.

Our observations in this study are consistent with observations made in murine neural stem cell populations [11]. Kalamakis *et al* reported an age-related drop in the total number and proliferative capacity of neural stem cells. In older animals, this neural stem cell population appears to be protected from full depletion by shunting cells into a quiescent state without increased senescence. Accordingly, the authors report that single-cell transcriptomics show minimal age-related changes [11]. The lack of alterations in transcriptomic profiles in both studies is unsurprising, since quiescence is thought to be controlled in large part through post-transcriptional mechanisms [9]. In addition to neural stem cells, age-related increases in quiescence have also been reported in skeletal muscle stem cells [19]. Brett et al report that skeletal muscle stem cells can be reactivated from this quiescent state to better participate in muscle regeneration after exercise. A common theme between these tissue types (muscle, neural, and lung) is that all three are relatively low turnover tissue with resident stem cell populations that are more likely to be quiescent unless responding to injury [8]. To enter quiescence, cells must exit the cell cycle, driven by upregulation of cyclin-dependent kinase (CDK) inhibitors and retinoblastoma protein (Rb), often controlled by extrinsic niche-derived signals. For example, quiescence may be reinforced by cadherin-mediated cell adhesion and contact inhibition [39]. Similarly, exit from quiescence may require alterations in niche-derived signals, including disruptions in physical contact that relieve contact inhibition [39]. It is possible that aging results in a dysregulation of this capacity to exit quiescence, consistent with reduced proliferation of highly proliferative cell populations in the brain, muscle and consistent with our data from the lungs. With similar age-related findings in all three of these highly proliferative cell populations, evidence is growing to support alterative aging strategies in these low turnover tissues that may provide superior protective mechanisms for these cell populations [8]. In turn, this may provide potential avenues for aging, tissue pathology and therapies in these tissues.

## Supporting information

**S1 File. Agyapong differential expression analysis table.**
(XLSX)

**S2 File. Agyapong et al data.**
(XLSX)

## Acknowledgments

The authors would like to acknowledge the contributions of Dr. Elizabeth Rozanski at the Cummings Tufts School of Veterinary Medicine in supplying canine lung tissue for this study, as well as numerous undergraduate research students in the Paxson lab that were also involved in this project, including Fernanda Perez-Alvarez, Azka Tanveer, Eric Carlson and Emily Bubonovich.

## Author contributions

**Conceptualization:** Nakesha Agyapong, Leslie Dominguez-Ortega, Brian Macdonough, Patrick Mulluso, Sagar Patel, Briti Prajapati, Brian Saville, Andrew Shapiro, Ethan Trim, Kara Battaglia, Jocelyn Herrera, Gianna Garifo-MacPartland, Dianne Newcomb, Latoya Okundaye, Heather Paglia, Julia Paxson.

**Data curation:** Nakesha Agyapong, Leslie Dominguez-Ortega, Brian Macdonough, Patrick Mulluso, Sagar Patel, Briti Prajapati, Brian Saville, Andrew Shapiro, Ethan Trim, Kara Battaglia, Jocelyn Herrera, Gianna Garifo-MacPartland, Dianne Newcomb, Latoya Okundaye, Heather Paglia, Julia Paxson.

**Formal analysis:** Nakesha Agyapong, Leslie Dominguez-Ortega, Brian Macdonough, Patrick Mulluso, Sagar Patel, Briti Prajapati, Brian Saville, Andrew Shapiro, Ethan Trim, Kara Battaglia, Jocelyn Herrera, Gianna Garifo-MacPartland, Dianne Newcomb, Latoya Okundaye, Heather Paglia, Julia Paxson.

**Funding acquisition:** Julia Paxson.

**Investigation:** Nakesha Agyapong, Leslie Dominguez-Ortega, Brian Macdonough, Patrick Mulluso, Sagar Patel, Briti Prajapati, Brian Saville, Andrew Shapiro, Ethan Trim, Kara Battaglia, Jocelyn Herrera, Gianna Garifo-MacPartland, Dianne Newcomb, Latoya Okundaye, Heather Paglia, Julia Paxson.

**Methodology:** Julia Paxson.

**Project administration:** Julia Paxson.

**Supervision:** Julia Paxson.

**Writing – original draft:** Nakesha Agyapong, Leslie Dominguez-Ortega, Brian Macdonough, Patrick Mulluso, Sagar Patel, Briti Prajapati, Brian Saville, Andrew Shapiro, Ethan Trim, Kara Battaglia, Jocelyn Herrera, Gianna Garifo-MacPartland, Latoya Okundaye, Heather Paglia, Julia Paxson.

**Writing – review & editing:** Nakesha Agyapong, Leslie Dominguez-Ortega, Brian Macdonough, Patrick Mulluso, Sagar Patel, Briti Prajapati, Brian Saville, Andrew Shapiro, Ethan Trim, Kara Battaglia, Jocelyn Herrera, Gianna Garifo-MacPartland, Dianne Newcomb, Latoya Okundaye, Heather Paglia, Julia Paxson.

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
