## [Decision Letter · Decision Letter 0]

PONE-D-25-06029Quiescence modulates age-related changes in the functional capacity of highly proliferative canine lung mesenchymal stromal cell populationsPLOS ONE

Dear Dr. Paxson,

Thank you for submitting your manuscript to PLOS ONE. After careful consideration, we feel that it has merit but does not fully meet PLOS ONE’s publication criteria as it currently stands. Therefore, we invite you to submit a revised version of the manuscript that addresses the points raised during the review process.

We look forward to receiving your revised manuscript.

Kind regards,

Nazmul Haque

Academic Editor

PLOS ONE

2. In your Methods section, please provide additional details regarding participant consent from the owners of the animals. In the ethics statement in the methods and online submission information, please ensure that you have specified (1) whether consent was informed and (2) what type you obtained (for instance, written or verbal). If the need for consent was waived by the ethics committee, please include this information.

 [Research reported in this study was supported by the National Institute on Aging, National Institutes of Health, grant number R15 AG064558-01 to JP.]. 

5.. Please review your reference list to ensure that it is complete and correct. If you have cited papers that have been retracted, please include the rationale for doing so in the manuscript text, or remove these references and replace them with relevant current references. Any changes to the reference list should be mentioned in the rebuttal letter that accompanies your revised manuscript. If you need to cite a retracted article, indicate the article’s retracted status in the References list and also include a citation and full reference for the retraction notice.

Additional Editor Comments (if provided):

Reviewers' comments:

Reviewer's Responses to Questions

**Comments to the Author**

1. Is the manuscript technically sound, and do the data support the conclusions?

Reviewer #1: Yes

Reviewer #2: Yes

2. Has the statistical analysis been performed appropriately and rigorously? 

Reviewer #1: Yes

Reviewer #2: Yes

3. Have the authors made all data underlying the findings in their manuscript fully available?

Reviewer #1: Yes

Reviewer #2: Yes

4. Is the manuscript presented in an intelligible fashion and written in standard English?

Reviewer #1: Yes

Reviewer #2: Yes

5. Review Comments to the Author

Reviewer #1: This study investigates whether age-related declines in canine lung mesenchymal stromal cells (LMSCs) are driven by senescence or a quiescent state. The authors do a good job of aligning their findings with a small but growing body of literature that challenges the traditional assumption that senescence is always the primary mechanism of proliferative decline in aging cells.

Comments:

1. Clarify how normality and equal variance assumptions were tested. If multiple t-tests were used, specify any methods (e.g., Bonferroni) to correct for multiple comparisons.

2. Provide numeric values (mean ± SD or SEM) alongside p-values for the main figures.

3. Briefly discuss known regulatory pathways of quiescence and how they might drive LMSC behavior with age.

4. Clarify if any sex- or breed-related differences appear in the data. While the study uses predicted lifespan to help account for breed size, acknowledging or ruling out sex/breed differences would strengthen the analysis.

Reviewer #2: accepted, Minimum Character Count Not Met.

Please use the space provided to explain your answers to the questions above. You may also include additional comments for the author, including concerns about dual publication, research ethics, or publication ethics. (Please upload your review as an attachment if it exceeds 20,000 characters) (Limit 200 to 20000 Characters)

6. PLOS authors have the option to publish the peer review history of their article (what does this mean? ). If published, this will include your full peer review and any attached files.

**Do you want your identity to be public for this peer review?** For information about this choice, including consent withdrawal, please see our Privacy Policy .

Reviewer #1: No

Reviewer #2: No

---

## [Author Response · Author response to Decision Letter 1]

13 May 2025

Dear Dr. Haque and Reviewers,

Thank you so much for your thoughtful review of this manuscript. We are very grateful for your comments and I believe that they will strengthen this study. Below are our responses to the comments from reviewer 1 (in green):

Reviewer #1: This study investigates whether age-related declines in canine lung mesenchymal stromal cells (LMSCs) are driven by senescence or a quiescent state. The authors do a good job of aligning their findings with a small but growing body of literature that challenges the traditional assumption that senescence is always the primary mechanism of proliferative decline in aging cells.

Comments:

1. Clarify how normality and equal variance assumptions were tested. If multiple t-tests were used, specify any methods (e.g., Bonferroni) to correct for multiple comparisons.

Normality was confirmed using the Shapiro-Wilk test. Equal variance assumptions were confirmed using F-tests. Please see lines 259-262 in the methods.

2. Provide numeric values (mean ± SD or SEM) alongside p-values for the main figures.

Done (please see relevant figure legends in the manuscript).

3. Briefly discuss known regulatory pathways of quiescence and how they might drive LMSC behavior with age.

Done. Please see lines 546-554 in the discussion.

4. Clarify if any sex- or breed-related differences appear in the data. While the study uses predicted lifespan to help account for breed size, acknowledging or ruling out sex/breed differences would strengthen the analysis.

Done. Please see lines 480-48 in the discussion.

---

## [Decision Letter · Decision Letter 1]

PONE-D-25-06029R1Quiescence modulates age-related changes in the functional capacity of highly proliferative canine lung mesenchymal stromal cell populationsPLOS ONE

Dear Dr. Paxson,

Thank you for submitting your manuscript to PLOS ONE. After careful consideration, we feel that it has merit but does not fully meet PLOS ONE’s publication criteria as it currently stands. Therefore, we invite you to submit a revised version of the manuscript that addresses the points raised during the review process.

We look forward to receiving your revised manuscript.

Kind regards,

Nazmul Haque

Academic Editor

PLOS ONE

Journal Requirements:

Reviewers' comments:

Reviewer's Responses to Questions

**Comments to the Author**

1. If the authors have adequately addressed your comments raised in a previous round of review and you feel that this manuscript is now acceptable for publication, you may indicate that here to bypass the “Comments to the Author” section, enter your conflict of interest statement in the “Confidential to Editor” section, and submit your "Accept" recommendation.

Reviewer #3: (No Response)

2. Is the manuscript technically sound, and do the data support the conclusions?

Reviewer #3: Yes

3. Has the statistical analysis been performed appropriately and rigorously? 

Reviewer #3: Yes

4. Have the authors made all data underlying the findings in their manuscript fully available?

Reviewer #3: Yes

5. Is the manuscript presented in an intelligible fashion and written in standard English?

Reviewer #3: Yes

6. Review Comments to the Author

Reviewer #3: The study investigates the senescence of mesenchymal stromal cells isolated from young and old donors in a dog model. The authors express the results considering cellular functions and proliferation capacity clearly and logically.

Comments:

1. Why was an animal model used in this study? Would it have been possible to conduct a clinical research? Can the results be applied clinically?

2. Can you clarify why mesenchymal stromal cells were derived from lung? Adipose derived mesenchymal stem cells would be easier to obtain.

3. There are 16 authors in this study. What was the contribution of each author to this work?

7. PLOS authors have the option to publish the peer review history of their article (what does this mean? ). If published, this will include your full peer review and any attached files.

**Do you want your identity to be public for this peer review?** For information about this choice, including consent withdrawal, please see our Privacy Policy .

Reviewer #3: No

---

## [Author Response · Author response to Decision Letter 2]

2 Jul 2025

Dear Dr. Haque and Reviewers,

Thank you so much for your thoughtful review of this manuscript. Below are our responses to the comments from reviewer #3:

Reviewer #3: The study investigates the senescence of mesenchymal stromal cells isolated from young and old donors in a dog model. The authors express the results considering cellular functions and proliferation capacity clearly and logically.

1. Why was an animal model used in this study? Would it have been possible to conduct a clinical research? Can the results be applied clinically?

An animal model was used in this study to facilitate ease of sample collection, to ensure access to donor age, and to explore MSC function across diverse animal species. A clinical research study would either require lung biopsies from patients, or a post-mortem sample collection system. Given the similarities between companion dog and human health-spans (see line 84-87 of the manuscript), we believe that the results from our study can be applied clinically.

2. Can you clarify why mesenchymal stromal cells were derived from lung? Adipose derived mesenchymal stem cells would be easier to obtain.

We were specifically interested in understanding how MSC function changed with donor age when isolated from a low turnover tissue system such as the lung that has clinically relevant increased pathology and chronic disease with age (see lines 37-41 in manuscript). In addition, in this study design, MSCs derived from post-mortem lung tissue are as easy to obtain as those derived from adipose tissue.

3. There are 16 authors in this study. What was the contribution of each author to this work?

The work in this study was performed by myself and teams of undergraduate research students (for a total of 16 researchers). Each team of students designed and optimized the assay that they were working on, analyzed the data, and contributed to writing the final manuscript. Because they were undergraduate students, each team worked on only one o the assays in the study, but all the authors listed contributed significantly to the study design, implementation, data interpretation, and manuscript writing.

---

## [Editor Report · Decision Letter 2]

Quiescence modulates age-related changes in the functional capacity of highly proliferative canine lung mesenchymal stromal cell populations

PONE-D-25-06029R2

Dear Dr. Paxson,

We’re pleased to inform you that your manuscript has been judged scientifically suitable for publication and will be formally accepted for publication once it meets all outstanding technical requirements.

Kind regards,

Nazmul Haque

Academic Editor

PLOS ONE
---

## [Editor Report · Acceptance letter]

PONE-D-25-06029R2

PLOS ONE

Dear Dr. Paxson,

I'm pleased to inform you that your manuscript has been deemed suitable for publication in PLOS ONE. Congratulations! Your manuscript is now being handed over to our production team.

Kind regards,

on behalf of

Dr. Nazmul Haque

Academic Editor

PLOS ONE